# Comprehensive Evolutionary Analysis of the *SMXL* Gene Family in Rosaceae: Further Insights into Its Origin, Expansion, Diversification, and Role in Regulating Pear Branching

**DOI:** 10.3390/ijms25052971

**Published:** 2024-03-04

**Authors:** Chunxiao Liu, Xianda Jiang, Susha Liu, Yilong Liu, Hui Li, Zhonghua Wang, Jialiang Kan, Qingsong Yang, Xiaogang Li

**Affiliations:** 1Jiangsu Key Laboratory for Horticultural Crop Genetic Improvement, Institute of Pomology, Jiangsu Academy of Agricultural Sciences, Nanjing 210014, China; lcx@jaas.ac.cn (C.L.); 20230986@jaas.ac.cn (S.L.); 20010005@jaas.ac.cn (H.L.); 20091204@jaas.ac.cn (Z.W.); 201800701@jaas.ac.cn (J.K.); 20030003@jaas.ac.cn (Q.Y.); 2College of Horticulture, Nanjing Agricultural University, Nanjing 210095, China; 2022804166@stu.njau.edu.cn; 3College of Horticulture and Landscape Architecture, Yangzhou University, Yangzhou 225009, China; mz120231498@stu.yzu.edu.cn

**Keywords:** genome-wide identification, gene expansion, pear branching, Rosaceae, *SMXLs*

## Abstract

*SMXL* genes constitute a conserved gene family that is ubiquitous in angiosperms and involved in regulating various plant processes, including branching, leaf elongation, and anthocyanin biosynthesis, but little is known about their molecular functions in pear branching. Here, we performed genome-wide identification and investigation of the *SMXL* genes in 16 angiosperms and analyzed their phylogenetics, structural features, conserved motifs, and expression patterns. In total, 121 *SMXLs* genes were identified and were classified into four groups. The number of non-redundant *SMXL* genes in each species varied from 3 (*Amborella trichopoda* Baill.) to 18 (*Glycine max* Merr.) and revealed clear gene expansion events over evolutionary history. All the *SMXL* genes showed conserved structures, containing no more than two introns. Three-dimensional protein structure prediction revealed distinct structures between but similar structures within groups. A quantitative real-time PCR analysis revealed different expressions of 10 *SMXL* genes from pear branching induced by fruit-thinning treatment. Overall, our study provides a comprehensive investigation of *SMXL* genes in the Rosaceae family, especially pear. The results offer a reference for understanding the evolutionary history of *SMXL* genes and provide excellent candidates for studying fruit tree branching regulation, and in facilitating pear pruning and planting strategies.

## 1. Introduction

Plants possess remarkable developmental plasticity and the adaptive ability to reshape their architecture in response to changes in light and other environmental signals or other artificial measures [1]. For example, branch development affects plant shape and, ultimately, crop yield and quality [2]. In fruit crops, tree pruning and shaping are essential to the promotion of branching, and understanding the molecular basis of branching is critical to fruit production [3]. Plant growth and development are regulated by many factors including hormone signaling, gene transcription, and others [4]. Strigolactones (SLs) are a family of terpenoid lactone hormones that control multiple developmental events in plants [4,5,6,7,8]. SLs were first identified in cotton root exudates as compounds that stimulate the germination of parasitic weeds [9] and play a critical negative role in regulating the growth and development of plant branches [10]. Later, SLs were shown to promote the symbiotic relationship between arbuscular mycorrhizal fungi and plant roots [11]. SLs are recognized as important plant endogenous hormones that are able to inhibit branch growth [4,5].

Similar to hormones such as auxins, gibberellins, and jasmonates, SL signaling requires proteasome-mediated repressor degradation to take place [10]. Recent studies have shown that SL signal transduction depends on interactions among several proteins, including SUPPRESSOR of MAX2 (MORE AXILLARY GROWTH2) 1-LIKE (SMXL), a key repressor in the SL pathway [12,13]. In the absence of SLs, SMXL6,7,8 proteins can perceive signals, form complexes that bind directly to the promoters of *SMXL6,7,8*, and repress their expression. Meanwhile, SMXL6,7,8 also serve as transcription factors. In the presence of SLs, DWARF14 (D14) binds to these compounds and promotes the formation of the SMXL6,7,8-D14-SCF-MAX2 complex, triggering the ubiquitin-mediated degradation of SMXL6,7,8, relieving transcriptional self-repression. SL-induced SMXL6,7,8 degradation causes transcriptional repression of SL-responsive genes such as *BRANCHED1*(*BRC1*), *Teosinte branched1/Cincinnata/Proliferating cell factor1* (*TCP1*), and *purple acid phosphatase1* (*PAP1*), which are essential for plant branching, leaf elongation, and anthocyanin biosynthesis, respectively. Newly synthesized SMXL proteins in turn repress transcription, forming a negative feedback loop that maintains the homeostasis of the SL pathway [14,15,16].

*SMXLs* constitute a conserved gene family that is ubiquitous in eukaryotic systems, from liverworts and mosses to green plants, and involved in regulating plant growth [17]. They have double well-conserved Clp-N motifs and P-loop NTPase domains that are unique to the nucleoside triphosphate hydrolase superfamily [13,18]. Additionally, two conserved and indispensable motifs, EAR (ethylene response factor-associated amphiphilic repression) and RGKT (Arg-Gly-Lys-Tr), which play an important role in SL signal transduction, have been identified in SMXL members [12,19,20,21]. The EAR motif can aid SMXL function but is not necessary [22]. The RGKT functional domain plays a key role in the perception of SL and the subsequent degradation of SMXL7/D53, which, when mutated or deleted, affects the degradation of SMXL repressors [12,13,19,23]. 

Recently, *SMXL* family genes have been studied comprehensively in various species such as *Arabidopsis thaliana* L. [24], apple [25], and poplar [26]. Individual members of this gene family are known to have unique functions [23,27,28,29]. Eight *SMXL* genes were identified and classified into four clades in *Arabidopsis*. *SMXL1,2* in clade I mainly regulate seed germination, hypocotyl length, and root and root hair development downstream of KAI2-mediated signaling [29]. *SMXL6,7,8* in clade II are mainly involved in SL signaling, the regulation of shoot branching, leaf morphology, and lateral root growth, and are reported to be the degradation targets of D14 [23,27,28,30]. However, *SMXL3* in clade III and *SMXL4,5* in clade IV do not respond to either karrikins (KARs) or SLs and are involved in phloem formation and primary root growth [29]. 

*Pyrus* L. (pear) is a genus in the Malinae subtribe of Rosaceae that includes European pear (*Pyrus communis* L.), Chinese white pear (*Pyrus bretschneideri* Rehder.), Japanese pear (*Pyrus pyrifolia* Burm.f.), *Pyrus ussuriensis* Maxim., and *Pyrus sinkiangensis* T.T.Yu. These fruits are grown world-wide and have been cultivated for >2500 years [31,32]. During pear cultivation, as branches do not germinate easily, bare branches can form, creating an empty tree crown, which ultimately affects fruit yield. Therefore, it is essential to promote the germination of dormant buds and trigger branching. 

To further the understanding of the evolution and function of *SMXLs* in angiosperms, especially their role in regulating branching in the Rosaceae, particularly in pear, we performed a genome-wide analysis of *SMXLs* on a large scale. In the current study, 16 representative angiosperm species were selected to show how the *121 SMXL* genes evolved and diverged from ancestral angiosperm *SMXL* gene lineages. We analyzed the classification, gene duplication events, structural features, conserved motifs, and phylogenetics of these *SMXL* genes, as well as their function in regulating pear branching. These results offer a reference for understanding the evolutionary history of *SMXL* genes and provide information for their use in future studies on the functional characterization of *SMXL* genes in branch regulation. The findings will also be useful in facilitating pear planting and pruning strategies.

## 2. Results

### 2.1. Identification and Classification of SMXL Genes 

To explore and increase our understanding of the evolution of SL signaling in the regulation of branching, we examined target repressor *SMXLs* in 16 angiosperm species, including 12 dicots (*A. thaliana*, *G. max*, *Ficus macrocarpa* L.f., *Solanum lycopersicum* L., *Vitis vinifera* L., and seven members of Rosaceae: *P. pyrifolia*, *Eriobotrya japonica* Lindl., *Gillenia trifoliate* Moench., *Prunus persica* Batsch., *Malus sylvestris* Mill., *Rosa chinensis* Jacq., and *Fragaria vesca* L.), two monocots (*Dioscorea alata* L. and *Oryza sativa* L.), and two basal angiosperms (*Nymphaea colorata* Verdc. and *A. trichopoda*). In total, 121 non-redundant *SMXL* gene family members were identified using a local BLAST method employing 8 SMXLs from *A. thaliana* as a query protein set. Among the 16 species investigated, the number of *SMXL* genes in each ranged from 3 to 18. *G. max* possessed the greatest number of *SMXL* genes (18), which was six-fold more than that in *A. trichopoda* (3). Each of the remaining 14 species contained 5 to 11 *SMXL* members. The open reading frame lengths of the SMXL proteins ranged from 722 to 1132 amino acids, with projected pIs of 5.76–8.82 and MWs of 80.67–112.53 kDa for the resulting proteins (Figure 1 and Appendix A). 

To investigate the classification of *SMXL* genes, we selected 11 species of the *rosids* lineage, including 7 Rosaceae, 1 Moraceae, 1 Fabaceae, 1 Brassicaceae, and 1 Vitaceae species. Additionally, we also selected one Solanaceae species of the asterids lineage, two monocots species, and two basal angiosperm species. The 16 species surveyed here occupy important phylogenetic locations, including three major angiosperm lineages and two basal angiosperms. The number of *SMXL* genes in each of the 16 species was as follows: *P. pyrifolia* (10), *M. sylvestris* (11), *E. japonica* (9), *G. trifoliate* (5), *P. persica* (6), *F. vesca* (5), *R. chinensis* (5), *F. microcarpa* (5), *G. max* (18), *A. thaliana* (8), *V. vinifera* (6), *S. lycopersicum* (7), *D. alata* (8), *O. sativa* (9), *N. colorata* (6), and *A. trichopoda* (3) (Figure 1).

The robust classification and phylogenetic relationships between the 121 *SMXL* genes in the 16 angiosperms (Figure 2a) clearly showed that the *SMXL* family genes were clustered into four branches comprising *SMXL1,2* (homologs of *SMAX1*, *SUPPRESSOR OF MAX2 1*, and *SMXL2*) as group I, *SMXL6,7,8* as group II, *SMXL3* as group III, and *SMXL4,5* as group IV (Figure 2a). These findings are consistent with previous classifications based on eight *SMXL* homologs in *A. thaliana*. Overall, group II contained the most *SMXL* family members (43), and group III contained the second highest number of *SMXL* genes (33), while groups I (22) and IV (23) contained the fewest and approximately equal numbers of *SMXLs*. To some extent, these results reflect *SMXL* gene duplication or loss events during evolution. *P. pyrifolia* contained two, four, two, and two *SMXL* genes in groups I, II, III, and IV, respectively, and we renamed them *PpySMXLs* (Appendix A) in accordance with their classification on the phylogenetic tree. Group IV was absent in *F. microcarpa* and *S. lycopersicum*, and group III was not found in *A. trichopoda*. The other 13 species contained all four groups of *SMXL* members (Figure 1 and Figure 2a). 

### 2.2. Gene Structures and Conserved Motif Composition of SMXL Genes

The intron–exon arrangement is often regarded as an important parameter in gene phylogenies. To confirm the classification of the 121 *SMXL* genes, we analyzed the predicted intron–exon organizations of their coding sequences. The results indicated that the genes had very conserved structures (Figure 2b). The majority of the *SMXL* family members in the four groups had two introns, with the exception only of *Ej00071280*, *PpySMXL8a*, *Gt00023314*, *Glyma.11G230700.1*, *Solyc06g051460.4.1*, and *AmTr_v1.0_scaffold00007.53* from group IV, *AtSMXL5* from group I, and *Glyma.10G121542.1* from group II, which had three introns (Figure 2b).

To further investigate the structural features of SMXL proteins, the MEME online tool was applied to explore the divergence of conserved motifs (Figure 2c). As a result, 40 conserved motifs in total were identified and renamed motifs 1–40 (Appendix A). As in previous reports, all SMXLs were characterized by a domain architecture of a double Clp-N motif and a P-loop containing a nucleoside triphosphate hydrolase [12,13]. Motif analysis revealed that almost all SMXL proteins had a highly conserved C-terminus and a variable N-terminus. Overall, the motifs in each group showed conserved organization, but some motifs were only contained in certain groups or SMXL members. For example, motifs 1–7, 10–12, 14–15, 17–18, and 22–23 were highly conserved in almost all of the SMXL family members (Figure 2c). Some motifs were conserved in one group, such as motifs 26, 31, 38, and 40, which were specific to group II; motifs 8, 16, 20, and 21, which were observed in groups I and II; motif 28, which was observed in groups II and IV; and motifs 34 and 36, which were contained in groups III and IV. Moreover, motifs 9 and 19 were absent from group IV, while motifs 13 and 19 were absent from group III, and motif 24 was absent from group II. Furthermore, some motifs dominated only in one group, such as motifs 27, 33, and 35 in group II; motifs 32 and 39 in group IV, and motif 25 in groups II and III. These different motif arrangements might indicate different gene functions. 

Taken together, our phylogenetic and intron–exon structure analyses of the *SMXL* genes in angiosperms clearly showed that there were four ancestral groups predating the divergence of monocots and dicots.

### 2.3. Synteny Analysis and Evolution of SMXL Genes

Whole-genome duplication (WGD) contributes to the generation of single-gene duplicates and has played important roles in plant genome function and evolution [33,34]. To explore the evolutionary mechanisms of the *SMXL* family, comparative genome mapping was performed to investigate the gene duplication events of *SMXL* genes in four Rosaceae members, including *P. Pyrifolia*, *E. japonica*, *G. trifoliate*, and *P. persica*. The results demonstrated a high level of conserved synteny among these species (Figure 3 and Appendix A). According to our results, more than six speculative *SMXL* genes were expected from the common ancestor of these four species. Previous studies have detected no recent WGD events in *P. persica* and *G. trifoliate* genomes [35,36]. Plant genomes are constantly in dynamic change and have suffered many gene losses since their two WGDs. There were six *SMXL* members in the *P. persica* genome but only five in *G. trifoliate*, so one *SMXL* gene in *P. persica* (*Prupe.2G071700.1*) had no corresponding link in *G. trifoliate*. Interestingly, one *P. persica* and one *G. trifoliate* gene recognized two *E. japonica* and *P. Pyrifolia* genes because of a recent WGD in the apple (Malus) tribe [37]. The number of *SMXL* genes in *E. japonica*, *M. sylvestris*, and *P. Pyrifolia* expanded rapidly, but through a long process of evolution and domestication, and after natural or artificial selection, there are now 10 *SMXL* genes in pear, 10 in *E. japonica* (including an incomplete short sequence), and 11 in *M. sylvestris*. 

To further trace the phylogenetic relationships between *SMXL* genes in angiosperms, we constructed an unrooted phylogenetic tree of the 121 *SMXL* genes from the 16 species (Figure 2a). This allowed us to obtain a clearer picture of the evolution of the *SMXL* genes. In the common ancestor of angiosperms, there are presumed to be four *SMXL* genes, with one member in each group. The *A. trichopoda* genome lost one member of group III during evolution, leaving three members. There was an ancient single WGD event that was probably shared among basal angiosperm (*N. colorata*) gene members [38]; now six *SMXL* genes remain and none of the four groups are absent (Figure 1 and Figure 2a). 

Studies have shown that a WGD event was shared by all grasses [39], and the *O. sativa* genome underwent an additional two WGD events accompanied by large-scale gene loss, with nine *SMXL* genes remaining. Another basal monocot, *D. alata*, contained eight *SMXL* genes. In eudicots, there was a shared γ paleohexaploidy event, noted as a whole-genome triplication (WGT) event [40]. The *S. lycopersicum* genome experienced one more WGT event, while *A. thaliana* and *G. max* experienced α and β WGD events, respectively, but they had different rates of gene loss [41,42,43], which contributed to the present number of *SMXL* genes in *A. thaliana* (8), *S. lycopersicum* (7), and *G. max* (18) (Figure 1 and Figure 2a). After Rosaceae split from Moraceae, a recent WGD event occurred in the apple tribe. Therefore, the number in the Maloideae is double that in the *G. trifoliate*, *P. persica*, *F. vesca*, and *R. chinensis* genomes. 

To further investigate the phylogeny and evolution of the *SMXL* gene family, an ML phylogenetic tree of pear *SMXL* genes was constructed and the 10 *SMXL* members were clearly classified into four groups (Figure 4). Furthermore, the spectrum of synonymous substitutions per synonymous site (*K*s) of the 10 *SMXLs* were confirmed to have two WGD peaks, one with a *K*s of ~0.13–0.21 and the other with a *K*s of ~1.66–1.93 (Appendix A). Additionally, the syntenic relationships between the 10 pear *SMXL* genes were constructed, and there were eight pair links (*PpySMXL1* vs. *PpySMXL2*, *PpySMXL4* vs. *PpySMXL5*, *PpySMXL6* vs. *PpySMXL7*, *PpySMXL6* vs. *PpySMXL8a*, *PpySMXL6* vs. *PpySMXL8b*, *PpySMXL7* vs. *PpySMXL8a*, *PpySMXL7* vs. *PpySMXL8b*, and *PpySMXL8a* vs. *PpySMXL8b*). Moreover, *PpySMXL3a* vs. *PpySMXL3b* might be a syntenic pair link but *PpySMXL3b* was unmapped on the chromosome (Figure 5). 

The broad-scale phylogenetic analyses suggested some subtree topologies that are consistent with the occurrence of ancient gene duplications. We conducted syntenic network analyses for *SMXL* genes using the collection of 16 available plant genomes (Figure 1). Syntenic *SMXL* genes were observed in all the selected plants. We visualized this subnetwork using Gephi [44] and color-coded the clusters using the k-clique percolation clustering method with k = 4 (Figure 6a). To reveal syntenic relationships between distant gene clades, we displayed pairwise syntenic relationships between the *SMXL* genes in a gene tree constructed for the entire gene family in the 16 genomes (Figure 6b). The results are consistent with previous classification and evolution analyses of the *SMXL* genes.

### 2.4. Secondary Structure Analysis and Prediction of Three-Dimensional Structure of SMXL Proteins

Three-dimensional structure prediction is critical to understanding protein function, so all of the 10 *PpySMXLs* proteins were included in the structure prediction analysis (Figure 7). The three-dimensional (3D) structure of the protein was predicted using Swiss-Model, and the quality of the model was evaluated via the global model quality estimate (GMQE). The SMXL protein is mainly composed of α-helices and random coils. We found that the GMQEs of the 10 selected PpySMXL proteins were all >0.57, indicating that the predicted 3D structures were reliable (Figure 7). Diverse 3D structures among SMXLs in different groups were observed, but the structures were shown to be similar between some members of the same group, suggesting that these proteins have various functions and conserved roles between duplicated genes. 

### 2.5. Expression Patterns of SMXL Genes in Regulating Pear Branching

Expression data were obtained from the transcriptome sequencing of *P. pyrifolia* 0, 1, 3, 5, 7, 14, and 21 days after fruit-thinning treatment to detect the genes involved in the bud germination and branching ability of pear. The 10 pear *SMXLs*, as well as 83 other genes associated with plant growth, cell division, and differentiation, and shoot apical meristem activity, were found to be differentially expressed at various time points. Most of the differentially expressed genes (DEGs) encoded transcription factors, including members of the ERF (ethylene responsive factor), ARF (auxin response factor), WRKY, TCP, SPL (SQUAMOSA promoter binding protein-like), and NAC (NAM, ATAF, and CUC2) families (Figure 8). Many of these genes were upregulated after fruit-thinning treatment, while others were downregulated. Most of the responsive genes were highly expressed at 7 days after treatment and then were downregulated from 14 days after treatment onward, which was consistent with the observations of buds germination. Of the 10 *SMXL* genes, *PpySMXL1*, *PpySMXL3a/b*, *PpySMXL4*, *PpySMXL5*, *PpySMXL6*, and *PpySMXL7* were more highly expressed after fruit-thinning treatment compared with the controls. However, *PpySMXL8a/b* exhibited repressed expression after the fruit-thinning treatment. In addition, other genes, especially transcription factor genes, which might be involved in branch development were upregulated after fruit-thinning treatment, (*NACs*, *ERFs*, *WRKYs*, and *MYBs*). For example, *NAC29* (*Ppy13g0816.1*), *NAC47* (*Ppy14g0042.1*), *NAC72* (*Ppy03g2089.1*), *WRK17* (*Ppy15g0899.1*), *WRK19* (*Ppy02g2370.1*), *WRK28* (*Ppy17g1273.1*), *WRK74* (*Ppy12g2041.1*), *ERF3* (*Ppy05g1753.1*), *ERF08* (*Ppy15g0661.1*), *ERF34* (*Ppy07g1095.1*), *TCP20* (*Ppy13g1302.1*), *ERF08* (*Ppy08g0767.1*), *ERF08* (*Ppy08g1027.1*), *ERF53* (*Ppy17g2333.1*), *ARFF* (*Ppy10g1359.1*), *WRK35* (*Ppy11g2580.1*) and *ERF08* (*Ppy15g0661.1*), were upregulated after fruit-thinning treatment. The reported genes downregulated by *SMXLs* that were suppressed following fruit-thinning treatment included *SPL12 (Ppy13g1277.1*), *SPL4* (*Ppy03g2160.1*), *TCP9* (*Ppy02g1731.1*), *SPL4* (*Ppy09g2030.1*), *TCP20* (*Ppy09g0071.1*), *TCP20* (*Ppy17g0069.1*), *SPL9* (*Ppy14g0522.1*), etc. (Figure 8).

We found the buds fully sprouted at 14 d after fruit-thinning treatment (Figure 9a), so we selected samples from 1 d, 5 d, and 14 d to verify the expression patterns. The expression of the *PpySMXLs* and the other 25 selected candidate DEGs was further verified via quantitative real-time (qRT)-PCR (Appendix A). The results were basically consistent with the RNA sequencing data. Most *PpySMXL* genes, including *PpySMXL1*, *PpySMXL3a/b*, *PpySMXL4*, *PpySMXL5*, *PpySMXL6*, and *PpySMXL7* were highly expressed after fruit-thinning treatment, particularly at 1 day after treatment, after which expression declined. The expression of *PpySMXL1*, *PpySMXL3a/b*, and *PpySMXL4* was higher than that in the controls until 14 days after treatment (Figure 9b). In particular, *PpySMXL3a/b* expression levels were 68 times higher than those in controls at 1 day after fruit-thinning treatment; however, the expression of *PpySMXL8a/b* was suppressed after fruit-thinning treatment (Figure 9b). The results indicated that members of the *SMXL* gene family play important roles in the regulation of pear branching after fruit thinning. The negative regulatory genes including *TCP* and *SPL* genes exhibited repressed expression after fruit-thinning treatment, which represented the start of branching (Figure 10).

## 3. Discussion

Since the discovery of D53/SMXL proteins in rice [12] and Arabidopsis [45], *SMXL* family members have gradually been characterized in many additional plant species. These proteins play important roles in various aspects of plant physiology, and evidence suggests distinct physiological roles for different *SMXL* orthologs [17]. However, the basis of their functional diversification has been largely unknown to date. We performed a genome-wide survey of *SMXL* genes in 16 plant species, including basal angiosperms, monocots, and dicots, and reconstructed the evolutionary history of this gene family. The *SMXL* gene family was conserved during evolution, and the genes can be grouped into four distinct clades in almost all of the 16 species studied, apart from *A. trichopoda*, *S. lycopersicum*, and *F. macrocarpa*, which were found not to contain group III, IV, and IV members, respectively (Figure 1). *SMXLs* are likely to have first appeared in liverworts and mosses, where they are involved in developmental responses [8]. The *SMXL* family likely arose from a single ancestral *SMXL* group through two whole-genome duplication events, at the point of seed plant and angiosperm origin [17,20]. Functionally, at present, the *SMXL* family can be divided neatly into a clade that mediates KAR/KL responses (SMAX1,2) and a clade that mediates SL responses (SMXL6,7,8) [46]. 

Phylogenetic and conserved motif analyses offered deep insights into the possible evolution and functional diversification of the *SMXL* homologs from ancient *SMAX1*/*SMXL1*. Previous studies indicated that *SMXL* homologs were completely absent in the lower plant groups [12,17]. In our study, *SMXL* genes were identified in all the 16 selected angiosperms, although some members were absent in certain species (Figure 1). Our evidence confirmed that the origin of the *SMXL* gene family predates the emergence of angiosperms, possibly in a common ancestor containing *SMAX1*/*SMXL1*. Previous studies based on genomic and RNA-seq data from several species belonging to bryophytes, lycophytes, and monilophytes concluded that these plants possess only one ancestral *SMXL* clade and, most often, a single *SMXL* copy [20]. Ancestral *SMXL* genes are the most similar to the SMAX1/*SMXL1* clade and are thought to be involved in the ancient KAR/KL response to SL [20,47]. In this study, a group that was the *SMAX1*/*SMXL1* clade of the *SMXL* gene family was present in all of the 16 species studied here (Figure 1 and Figure 2), indicating that these genes were conserved during evolution.

WGD or polyploidy contributes to the generation of single-gene duplicates or whole gene families and has played crucial roles in plant genome function and evolution [33,34]. Instead of fungi and animals, the most frequent occurrence of paleo-polyploidization has been detected in angiosperms [34]. In the present study, we found an obvious expansion of the *SMXL* gene family in angiosperms, starting from 3–4 members in basal angiosperms to 9–11 members in the Maleae (Figure 1). This expansion of *SMXL* genes occurred thanks to a WGD in the common ancestor of *P. persica*, *M. sylvestris*, and *G. trifoliate*. After several rounds of WGD events, the number of genes in the *SMXL* family in pear reached its current number. 

The common ancestor of mesangiosperms underwent WGD event(s), while basal angiosperms (*N. colorata* and *A. trichopoda*) did not. Consequently, the number of *SMXL* genes doubled, but then homologs were lost. Interestingly, the gene loss events occurred selectively in groups I, III, and IV (Figure 1). When the common ancestor of pear, apple, and loquat underwent another recent WGD event [37], the numbers of *SMXL* genes doubled again (Figure 1). We constructed a phylogenetic tree of the 10 pear *SMXL* genes, with the results clearly showing expansion of these genes (Figure 4). Furthermore, the spectrum of the *K*s values of the 10 *SMXLs* confirmed two WGD peaks, one with a *K*s ~0.13–0.21 that occurred approximately ~7.0–11.3 million years ago (MYA) and the other with a *K*s ~1.66–1.93 that took place ~89.6–104.2 MYA (Appendix A). These findings demonstrated the evolution and expansion of *SMXL* genes.

*SMXL* genes regulate diverse aspects of plant development and responses to various environmental signals in the SL signaling pathway [10,24]. In *Arabidopsis*, *AtSMAX1* responds to KAR signals to regulate seed germination and hypocotyl length [17,48]. AtMAX2 is responsible for the poly-ubiquitination of target SMXL proteins, which are consequently degraded by the 26S proteasome, resulting in downstream signaling. *AtSMXL3,4,5* do not respond to KARs or SLs [23], and their involvement in primary phloem formation has been discovered relatively recently [29]. AtSMXL6,7,8 respond to SL signals to inhibit the expression of transcription factors BRC1, TCP1, and PAP1 by binding to the transcriptional corepressor protein TPL and TPL-related protein (TPR), thereby regulating plant branching, leaf morphology, and lateral root growth [27,30]. In addition, it has been found that SMXL6,7,8 are not only repressors but also transcription factors, which negatively regulate their own transcriptional expression to maintain the dynamic balance of SMXL6,7,8 proteins and SL signal responses [24]. There are studies showing that the expression levels of *SMXL6,7* are also regulated by the light environment [49]. 

Our previous study revealed that fruit-thinning treatment could induce bud germination and that *SMXL* gene members respond to the germination process after fruit-thinning treatment. In this study, we detected transcripts of all 10 *PpySMXL* genes in pear after fruit-thinning treatment and observed their distinct expression levels (Figure 8). The expression of *PpySMXL6,7,8a/b* were the highest, followed by *PpySMXL1* and *PpySMXL2* (with no difference between the experimental group and control), while the expression levels of *PpySMXL3,4,5* were the lowest (Figure 8). Using qRT-PCR, we detected the relative expression levels of *PpySMXL1*, *PpySMXL3a/b*, *PpySMXL4*, *PpySMXL5*, *PpySMXL6*, *PpySMXL7*, and *PpySMXL8a/b*, which were consistent with those found in previous studies of model plants but showed a unique regulation of the pear bud after fruit-thinning treatment (Figure 9). This regulatory function might be related to the expansion, functional redundancy, and functional differentiation of *SMXL* genes in pear. With respect to evolution, in the short term, the functions of homologous *SMXL* genes in pear are relatively undifferentiated, and redundant homologous genes may play similar roles, but they may have more functions in the future [50]. However, the genes have their own characteristics and functional responses to fruit-thinning treatment. For example, *PpySMXL5,6,7* were found to respond quickly at an early stage (1 day) after treatment before being downregulated (Figure 9b), suggesting an important role in the early fruit thinning response. Previous studies have reported that *PpySMXL6,7,8* are involved in regulating shoot branching, leaf elongation, and anthocyanin biosynthesis [24]. In this study, the expression of *PpySMXL6* and *PpySMXL7* was significantly upregulated after fruit-thinning treatment (Figure 9b), but *PpySMXL8* was inhibited (Figure 9b), indicating the probable functional differentiation of these *SMXL* genes in pear. Other new functions in other signaling pathways involved in regulating pear growth and development may also be present, although to date, these have not been identified. In summary, *SMXL* genes involved in the regulation of branching in pear are candidates for the study of branching in plants.

Besides the *SMXL* genes, there are also some other genes associated with cell division, differentiation, and SAM activity, the responses of which to branching were identified, including transcription factors belonging to the *ERF*, *SPL*, *TCP*, *WRKY*, and *NAC* families, among others (Figure 8). We selected and detected the expression of 25 genes in response to fruit-thinning treatment here (Figure 10). Transcription factors play a vital role in plant development and the regulation of gene expression, forming a complex gene regulatory network [51]. Most of the selected transcription factors were highly expressed under fruit-thinning treatment in the regulation of branching traits. The TCP transcription factors are involved in the growth of lateral meristems, cell proliferation, and the regulation of hormones [52,53]. In peach, *PpTCP18* can reduce secondary branches through the brassinolide pathway [54]. However, we did not find similar results in this study might because we missed the closest homologs of *TCP18*. The *SPL* genes are mainly expressed in meristems and are critical to regulating the branching and vegetative growth of alfalfa plants [55]. Previous studies indicated that overexpression of *SPL13* inhibits the growth of axillary buds and reduces the number of lateral branches [56]. We found in this study that the *SPL4* and *SPL9* genes were significantly downregulated after the fruit-thinning treatment (Figure 10). The results show that *SPL* genes were indeed involved in tree branching, and that this is a complex regulatory process involving a large number of genes. Furthermore, there are many other transcription factor genes including *ERF*, *NAC*, *WRKY*, *MYB*, *WOX*, *CYP*, and so on that were induced at more than one time point via fruit-thinning treatment (Figure 10). The present research not only provides good candidate genes for fruit tree branching research, but also provides a new perspective for researchers studying plant branching.

## 4. Materials and Methods

### 4.1. Data Collection

The characteristics and phylogenetic relationships between *SMXL* genes from 16 representative species with relatively complete annotated genome data were investigated. In all the 16 species, we selected 11 species from the rosids lineage, including 7 Rosaceae (*P. pyrifolia*, *P. persica*, *F. vesca*, *M. sylvestris*, *E. japonica*, *R. chinensis*, and *G. trifoliate*), 1 Moraceae (*F. microcarpa*), 1 Fabaceae (*G. max*), 1 Brassicaceae (*A. thaliana*), and 1 Vitaceae (*V. vinifera*) species. Additionally, we also selected one *Solanaceae* species (*S. lycopersicum*) from the asterids lineage, two monocot sspecies (*D. alata* and *O. sativa*), and two basal angiosperm species (*N. colorata* and *A. trichopoda*). Genomic sequences, gene annotations, and gene models were obtained from the following databases: Phytozome (https://phytozome-next.jgi.doe.gov/, accessed on 5 August 2023) (*A. thaliana*, *P. persica*, *F. vesca*, *G. max*, *D. alata*, *O. sativa*, and *A. trichopoda*); Genome Database for Rosaceae (GDR: https://www.rosaceae.org/, accessed on 5 August 2023) (*P. pyrifolia* and *M. sylvestris*); Solanaceae Genomics Network (https://solgenomics.net/, accessed on 5 August 2023) (*S. lycopersicum*); Grapedia (https://grapedia.org/genomes/, accessed on 6 August 2023) (*V. vinifera*); Institut national de la recherche agronomique (https://lipm-browsers.toulouse.inra.fr/pub/RchiOBHm-V2/, accessed on 6 August 2023) (*R. chinensis*); Genome Warehouse GWH: https://ngdc.cncb.ac.cn/, accessed on 6 August 2023) (*F. microcarpa* and *N. colorata*); and China National GeneBank (CNGB: https://db.cngb.org/search/, accessed on 6 August 2023) (*E. japonica* and *G. trifoliate*). 

### 4.2. SMXL Gene Identification and Classification

Eight previously identified and characterized SMXL proteins from *A. thaliana* (GenBank: SMAX1, AT5G57710.1; SMXL2, AT4G30350.1; SMXL3, AT3G52490.1; SMXL4, AT4G29920.1; SMXL5, AT5G57130.1; SMXL6, AT1G07200.2; SMXL7, AT2G29970.1; SMXL8, AT2G40130.2) were employed to construct a set of query proteins [57]. The candidate SMXL proteins in the other 15 species were identified via a local BLAST search using TBtools V1.113 [58] with Expect values below 1 × 10^−10^. Redundant and short sequences (<100 amino acids) were filtered out manually. Subsequently, abnormal sequences were deleted manually. The biochemistry of each SMXL protein, including the number of amino acids, isoelectric point (pI), and molecular weight (MW), was determined using online ExPASy software (https://www.expasy.org/, accessed on 16 August 2023).

### 4.3. Phylogenetic Tree, Gene, and Three-Dimensional Protein Structures, Conserved Domains, and Motifs 

Multiple-SMXL-sequences alignment was performed using MEGA11.0 [59], and the resulting sequence was trimmed using the Simple MSA Trimmer assembled in TBtools V1.113 [58]. The phylogenetic tree of species was constructed in accordance with the AGP IV system [60], and polyploidization events described in previous studies [37,40,61,62,63] are mapped onto the tree. The phylogenetic tree with 121 *SMXL* genes in 16 angiosperms was created based on the maximum likelihood method and Poisson’s correction model using 1000 bootstrap replicates [64], and then edited and visualized using FigTree software (http://tree.bio.ed.ac.uk, accessed on 12 September 2023). *SMXL* gene structure information was parsed from the GFF files of each genome. Gene Structure View (Advanced) within TBtools V1.113 was used to create diagrams of domain locations and gene structures. MEME online tools (https://meme-suite.org/meme/tools/meme, accessed on accessed on 21 September 2023) were used to predict conserved motifs using the following parameters: repeats per sequence (any); motif width range (6–50 amino acids); and maximum number of motifs (50). The tertiary structures of SMXL proteins were modeled and displayed using the Swiss-Model interactive tool (https://swissmodel.expasy.org/interactive, accessed on 2 September 2023).

### 4.4. Genome Synteny and Gene Duplication Analysis

Genome assembly and annotation files were obtained from Genome Database for Rosaceae (*P. pyrifolia*); Phytozome (*P. persica*) and China National GeneBank (*E. japonica* and *G. trifoliate*). Genome synteny information for *P. pyrifolia*, *P. persica*, *E. japonica*, and *G. trifoliate* was calculated using MCScanX [65]. MCScanX was also used to recognize whole-genome duplication events in *SMXL* genes from pear (*P. pyrifolia*). 

*SMXL* syntenic networks were analyzed with CFinder v2.0.6 [66] using the unweighted CPM algorithm and no time limit. All possible k-clique communities for the *SMXL* gene syntenic networks were identified using k = 4 as the clique community threshold—in this scenario, one *SMXL* gene (node) involved in a subnetwork community needed to have at least two connections (edges) with other nodes in the community. Increasing k values make the communities smaller and more disintegrated but also more connected. TBtools V1.113 was applied to perform a collinearity analysis of *SMXL* genes with default parameters. The *K*s values of syntenic genes were calculated using *K*a*K*s_Calculator 3 (https://ngdc.cncb.ac.cn, accessed on 15 September 2023) [49].

### 4.5. Plant Materials and Treatment

Cultivar ‘Qiuyue’ (*P. pyrifolia*), a widely planted cultivar imported from Japan, was used for fruit-thinning treatments in this study. It was grown in the farm. Samples of buds were collected at 0, 1, 3, 5, 7, 14, and 21 days after fruit-thinning treatment and immediately frozen using liquid nitrogen for future experiments. Each treatment had three biological replicates. The frozen samples were stored at −80 °C until further use. 

### 4.6. Expression Analysis of Pear SMXL Genes

Expression data for *SMXL* genes were obtained from unpublished transcriptome data collected following a fruit-thinning treatment to detect the ability of pear branches to germinate. HeatMap Illustrator within TBtools V1.113 [58] was used to draw a graphic representation of expression patterns. Gene expression levels at the different time points were calculated in accordance with the log2 FPKM values. Based on heatmap results, the 10 *PpySMXLs* and 25 candidate genes for fruit-thinning treatment were selected to verify the RNA-seq data.

Total RNA was isolated using a plant RNA purification kit (Takara, Japan) from leaf tissues in accordance with the manufacturer’s instructions. Then, gDNA digestion was carried out using 3 μL 5 × gDNA Digester Mix and RNase-free ddH_2_O. The PCR products were sequenced to verify the absence of gDNA in the RNA fraction. The pure RNA was reverse-transcribed into cDNA using the reverse transcription kit (Vazyme, Nanjing, China). The specific primers were designed based on the selected gene sequences using TBtools V1.113 [58], and then, all the qRT-PCR products were sequenced to select the specific primers for follow-up detection. The expression of the selected genes was analyzed using a BIOER, FQD-96A real-time PCR system with 2×HS Taq Universal SYBR Green qPCR Master Mix, SAIPUBIO. Gene-specific primers were designed based on the selected gene sequences using TBtools V1.113 [58]. Actin from *P. pyrifolia* (*Ppy01g0117.1*) was used as the reference gene. The amplification parameters were as follows: 95 °C hold for 10 min, followed by 40 cycles at 95 °C for 15 s, 58 °C for 15 s, and 72 °C for 15 s. For the melting curve stage, the default settings were chosen. Monospecific products were identified by inspecting the melting curves (Appendix A). The relative expression level for each gene was calculated with the 2^−ΔΔCt^ method. Each sample analysis was repeated three times. A *T*-test was used for statistical analysis. All the primers used in this paper are listed in Appendix A.

## 5. Conclusions

In summary, this study provides important insights into the *SMXL* gene family in Rosaceae species and their roles in regulating pear branching. In conclusion, 10 *PpySMXL* genes were identified in a pear genome, and 121 *SMXL* genes were comprehensively collected from 16 representative species. They were phylogenetically divided into four clades and revealed clear gene expansion events over evolutionary history. A genomic collinearity analysis provided evidence that *SMXLs* have undergone several rounds of WGD events over evolutionary history that have led to their current expansion. However, the number varied from 3 to 18 in different angiosperms. According to the protein structure and motif analysis, all SMXL proteins are structurally very conserved. We also analyzed the 3D structure, phylogenetic relationships, exon–intron arrangement, gene duplication, and induced expression patterns of these genes’ response to fruit thinning in regulating pear branching. Our findings provide excellent candidate genes and shed light on the significant roles of the *SMXL* gene family in regulating tree branching and molecular breeding, and in facilitating pear planting strategies, as well as providing a reference for studies of fruit tree pruning and production.

## Figures and Tables

**Figure 1 ijms-25-02971-f001:**
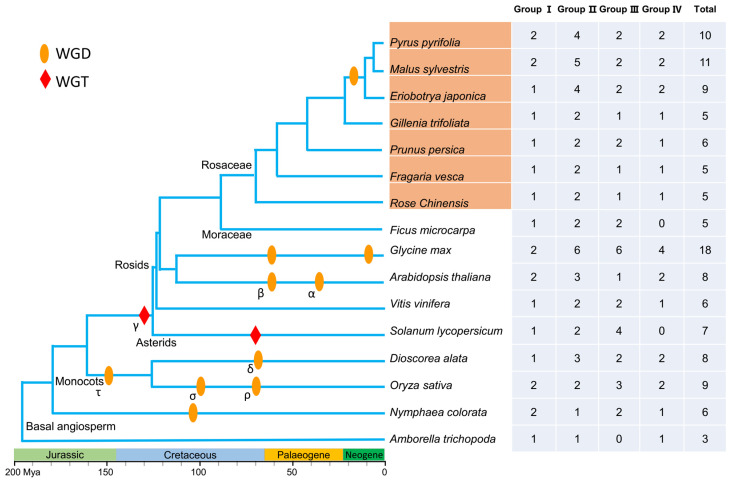
Identification and classification of *SMXL* genes in 16 angiosperm species. The phylogenetic tree was constructed in accordance with the AGP IV system. The estimated time is from GSA Geologic Time Scale (https://www.geosociety.org/GSA/GSA/timescale/home.aspx, accessed on 10 August 2023). Polyploidization events described in previous studies mentioned in the method are mapped onto the tree (orange ovals and red diamonds). The total number of *SMXL* genes and their classification in each plant are shown in the figure. WGD: whole-genome duplication; WGT: whole-genome triplication; Mya: million years ago.

**Figure 2 ijms-25-02971-f002:**
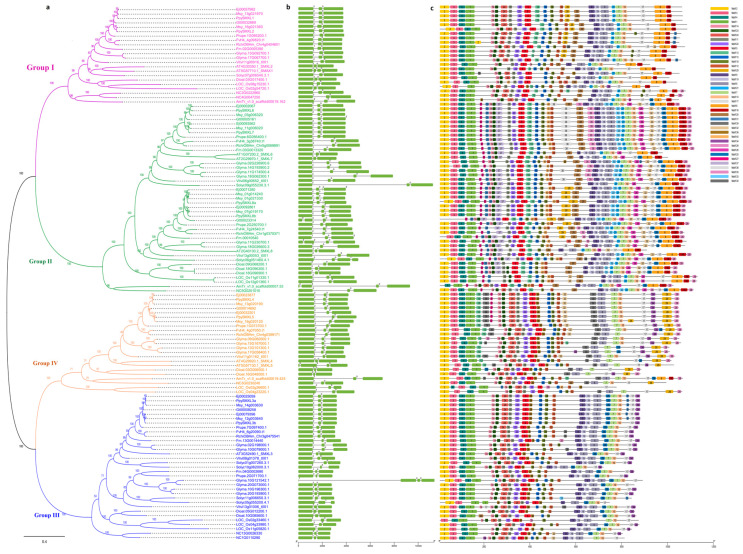
Phylogenetic tree analysis, exon/intron structures, and motif distribution of *SMXL* genes. (**a**) The phylogenetic tree of 121 *SMXL* genes in 16 angiosperm species constructed using the maximum likelihood (ML) method and Poisson’s correction model. Four groups are represented in variously colored branches. (**b**) The distribution of the intron and exon organization of *SMXL* genes shown using TBtools V1.113 software. The green boxes indicate the CDSs (coding sequences) and the gray lines indicate the introns. The scale bar at the bottom allows for the estimation of the length and relative position of each box. (**c**) The conserved motifs of SMXL proteins. Differently colored boxes indicate various motifs.

**Figure 3 ijms-25-02971-f003:**
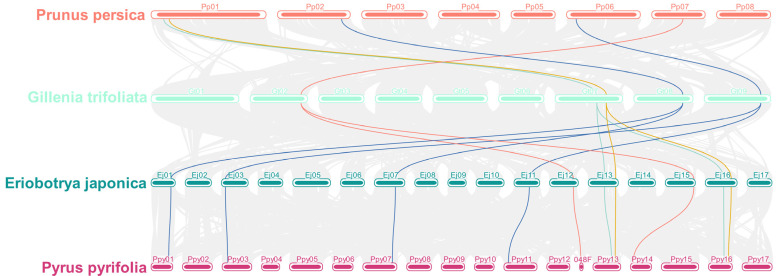
Synteny analysis of *SMXL* genes among *P. pyrifolia*, *E. japonica*, *G. trifoliate*, and *P. persica*. The gray lines in the background indicate all synteny blocks within the *P. pyrifolia* genome and the other three genomes, and blue lines indicate the duplicated *SMXL* gene pairs among them.

**Figure 4 ijms-25-02971-f004:**
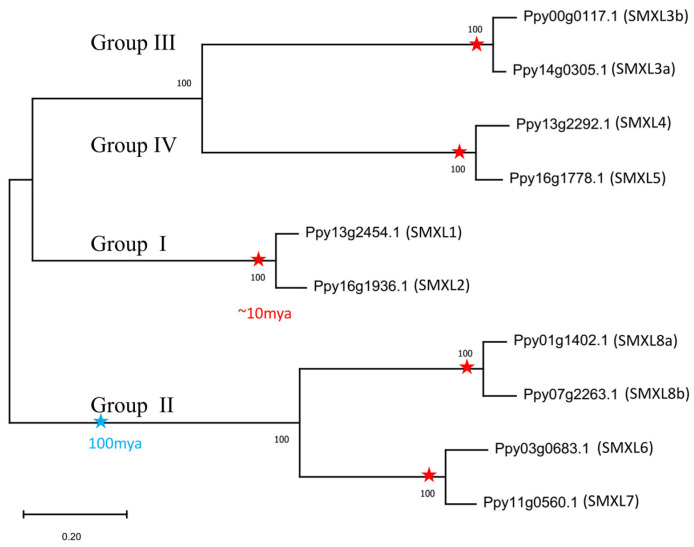
Phylogenetic tree of *SMXL* genes in pear. The ML tree was constructed with MEGA 11.0 software using amino acid sequences based on the WAG model; the numbers on the branches represent the bootstrap supports. *SMXL* genes are from *P. pyrifolia*. We calculated the *K*s value of homologous gene pairs in each subfamily (Appendix A), and then interpreted the replication nodes based on the *K*s value. The common ancestor nodes marked by red stars represent the duplication event during the recent pear lineage-specific WGD. The common ancestor nodes marked by blue stars represent the duplication event during the ancient legume WGD.

**Figure 5 ijms-25-02971-f005:**
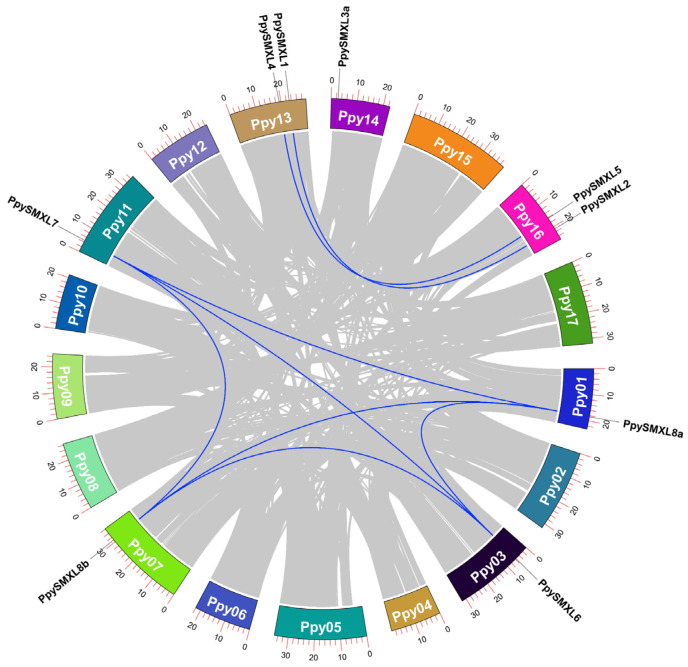
Schematic representations for chromosomal distribution and interchromosomal relationships of pear *SMXL* genes. Differently colored lines indicate all synteny blocks in the pear genome. The blue lines indicate a gene pair that is duplicated with *SMXL* genes. The chromosome number is indicated in each chromosome bar. The lengths of the chromosomes are marked using scales, with each small scale indicating 500,000 amino acids, and every 10th scale is marked using a number.

**Figure 6 ijms-25-02971-f006:**
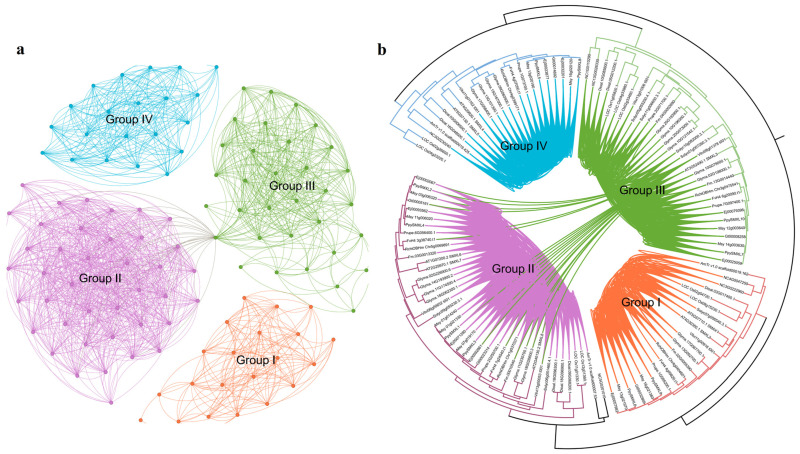
Synteny network of *SMXL* family genes and syntenic relationships within and between three groups of genes. (**a**) The synteny network of *SMXL* family genes. Communities were rendered based on the clique percolation method at k = 4. The size of each node indicates the number of connected edges (the node decree). The communities are denoted by the four groups (Group I–IV) involved. (**b**) Syntenic relationships among the *SMXL* genes within the phylogenetic tree. Each connecting line located inside the inverted circular gene tree indicates a syntenic relationship between two *SMXL* genes. Lineage information is contained in the branches.

**Figure 7 ijms-25-02971-f007:**
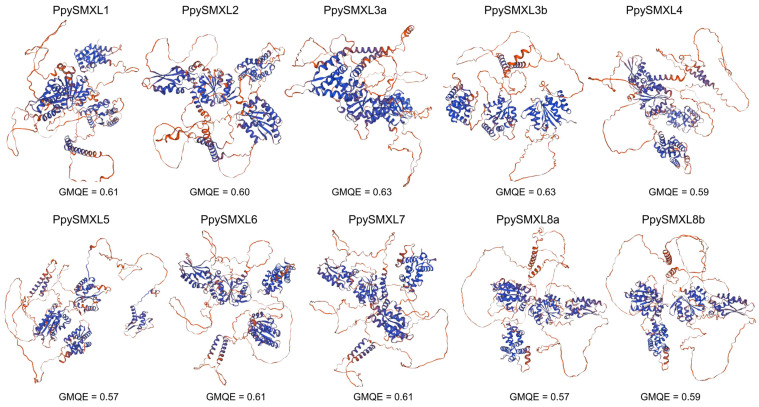
The predicted three-dimensional (3D) structure of nine PpySMXL proteins. Using the protein homology modeling method based on the SMXL structure of the Swiss-Model database, the structure with the highest score was chosen as the optimal structure for the PpySMXL protein.

**Figure 8 ijms-25-02971-f008:**
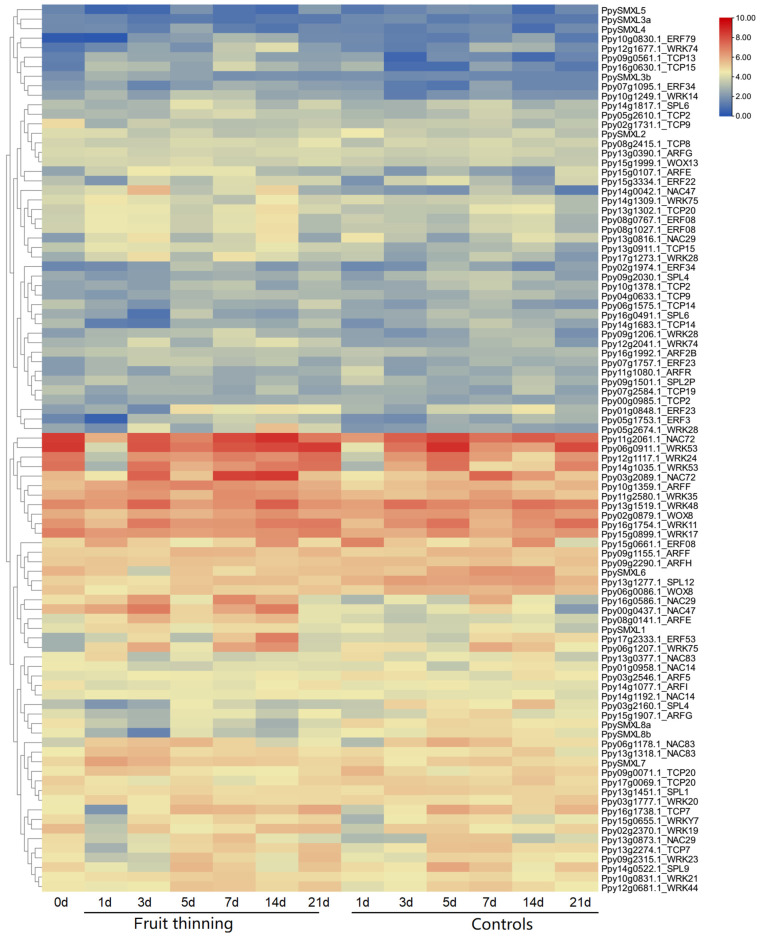
Expression pattern of *PpySMXL* genes under fruit-thinning treatment at different stages. The transcript levels of *PpySMXL* genes at 0 d, 1 d, 3 d, 5 d, 7 d, 14 d, and 21 d were investigated based on transcriptome data. The expression level of *PpySMXL* genes is shown as a heatmap of log2 (FPKM) values. The color scale shows increasing expression levels from blue to red.

**Figure 9 ijms-25-02971-f009:**
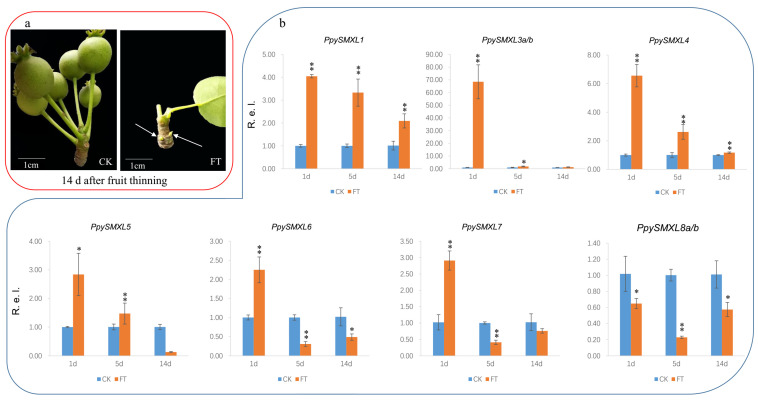
(**a**) Bud development after 14 days of fruit thinning (FT) treatment; blank treatments act as controls (CK). (**b**) Gene expressions of *SMXL* genes under fruit-thinning treatment determined via qRT-PCR. The mean expression values were calculated from three independent replicates. The X-ray indicates the number of days after treatment. Mean values and standard errors were calculated from three replicates. A *T*-test was used for statistical analysis. The asterisk and double asterisks represent significant differences at the levels of 0.05 and 0.01, respectively. R.e.l indicates the relative expression level.

**Figure 10 ijms-25-02971-f010:**
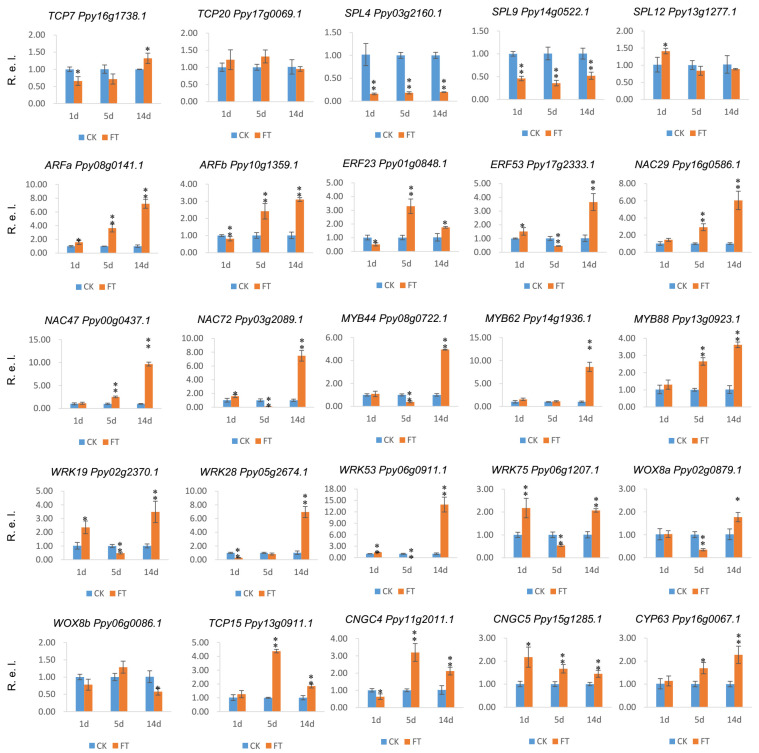
Gene expressions of selected genes under fruit-thinning (FT) treatment determined via qRT-PCR, with blank treatments as controls (CK). The mean expression values were calculated from three independent replicates. The X-ray indicates the number of days after treatment. Mean values and standard errors were calculated from three replicates. A *t*-test was used for statistical analysis. The asterisk and double asterisks represent significant differences at the levels of 0.05 and 0.01, respectively. R.e.l indicates the relative expression level.

## Data Availability

The data presented in this study are available on request from the corresponding author. The data are not publicly available due to another unpublished manuscript.

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
