# Peer review of "Comprehensive Evolutionary Analysis of the SMXL Gene Family in Rosaceae: Further Insights into Its Origin, Expansion, Diversification, and Role in Regulating Pear Branching"

_ijms, 2024, doi:10.3390/ijms25052971_

Round 1

Reviewer 1 Report

Comments and Suggestions for Authors

Dear Authors

The manuscript is well written and presented.  An addition of a pathway or a gene regulatory network highlighting the occurrence or the role of SMXL genes in branching can be included. No single gene is responsible for branching and indicating only SMXL in the realtime qRT-PCR is somewhat old fashioned though tissue specific expression is clearly observed. The discussion should deliberate on this aspect.

Best

Comments on the Quality of English Language

Minor editing is required

Reviewer 2 Report

Comments and Suggestions for Authors

The article entitled “Comprehensive Evolutionary Analysis of the SMXL Gene Family in Rosaceae: Further Insights into its Origin, Expansion, 3Diversification and Role in Regulating Pear Branching” is devoted to the study of the classification, gene duplication events, structural features, conserved motifs, and phylogenetics of these SMXL genes and their function in regulating pear branching. These results obtained during the study may be used for the understanding of the SMXL genes evolutionary history and provide information for use in future studies on the functional characterization of SMXL genes in branch regulation. The applied importance of the study lays in the field of pear planting and pruning strategies. The manuscript is well written and illustrated.

The main question to the authors is about qPCR parameters and strategies. The RNA isolation for following cDNA synthesis and qPCR analyses is a very critical in term of RNA quality and absence of gDNA contamination. If RNA is contaminated by gDNA, the results of qPCR will be wrong due to specific primers binding to the gDNA. Authors should provide a method of gDNA digestion, and the results of PCR with isolated RNA and specific primers to prove the absence of gDNA in the RNA fraction. Or at least write about this in Methods part. What was the qPCR primers efficiency? It is important to be sure that qPCR primers binding is specific. If the estimated PCR product size is matching with the obtained it doesn’t mean that you amplified the right target. Only sequencing of PCR product may prove the specificity of primers binding. The same applies to the reference genes. All PCR products should be isolated and sequenced to prove specificity of primers binding. If sequencing of PCR product was done, please indicate this in the text.

Authors write, that “Nonspecific products were identified by inspecting melting curves.”. Why did you have unspecific products? qPCR primers should be designed to exclude any unspecific products.

The writing of SMXL through the entire text should be unified (in Italic or not in Italic type).

Also the same problem with Italic type for different species and family names should be solved.

L71: “Arabidopsis” – change to the Italic type.

L78: “Rosaceae” – should be not in Italic type, like in headline. Same for the line 101.

L100: “G. max” – please write full name, like for other species.

L101: “P. pyrifolia” - please write full name, like for other species.

L108 “A. trichopoda” - please write full name, like for other species.

L119: Plant Family names usually are written in not Italic. Please check this for the whole text.

L148: Please check space between words.

L184-185: Please check all the scientific names for Italic type.

L192: “G. trifoliate” – change to Italic type.

Figure 3: Species names at the picture should be written in Italic type. Please change.

L199: "P. pyrifolia” – change to Italic type.

L219-210: “O. sativa” – change to Italic type.

L211: “D. alata” – change to Italic type.

L214: “A. thaliana” – change to Italic type.

L216: “A. thaliana” – change to Italic type.

L217: Family names should be not in Italic.

L279: “P. pyrifolia” – change to Italic type.

L373: “Maleae” - should be not in Italic.

L389-L402: Please check the gene names. Should be in Italic like in whole text.

L422-L430: Please check the gene names. Should be in Italic like in whole text.

L435: “Rosaceae” – should be not in Italic type.

L436: “Moraceae” – should be not in Italic type.

L438: “Solanaceae” – should be not in Italic type.

L444: “Solanaceae” – should be not in Italic type.

L451: A. thaliana - change to Italic type.

L479-483: Please check the gene names. Should be in Italic like in whole text.

L494: P. pyrifolia - change to Italic type.

Reviewer 3 Report

Comments and Suggestions for Authors

The issues of regulating the development of plants and its individual organs are extremely important. All regulatory mechanisms are based on the formation of a negative feedback loop. One pathway regulates the activation of a certain process; there must be another that inhibits this process. Identifying the participants in the process of branching regulation seems interesting not only from a scientific point of view, but also from a practical one. Clarification of the branching process can help increase the yield of fruit plants. 

30- this sentence is inappropriate, since the study is not devoted to the impact of environmental changes on architecture

49 - what is this class of D14 proteins?

Figure 1 - indicate where you got the data that there were 200 mya?

133 -gene indicate in italics and further in the text

Figure 2 - CDs?, 148- remove space

162 - references according to the rules

220-ML phylogenetic tree?

All plant names in italics (especially in the materials and methods section)

242 - phrase is unclear

Figure 8 - on the X-axis are 1d, 3d, etc.?

The data in Figure 8 are shown only after thinning; there is no data before this procedure (O control). Therefore they cannot be compared.

339- D53/SMXL is a different protein (D14) and a larger family of proteins than SMXL and how are they fundamentally different?

353 - SMAX or SMXL

references are checked and prepared at the request of the journal

Reviewer 4 Report

Comments and Suggestions for Authors

In this comprehensive study, the authors identified and analyzed the SMXL gene family in 16 angiosperms, shedding light on their molecular functions in pear branching. The genome-wide investigation, including phylogenetic analysis, structural features, conserved motifs, and expression patterns, revealed a total of 16,121 SMXL genes classified into four groups. The study provides valuable insights into the evolutionary history of SMXL genes, highlighting their potential role in regulating plant processes and offering a reference for understanding fruit tree branching, especially in the context of pear pruning and planting strategies. Overall, the manuscript is quite interesting and valuable, although it requires some improvements before publication.

§  Include the initials of the authors when mentioning the species for the first time, e.g. Amborella trichopoda Baill.

§  Keywords should be arranged alphabetically and not repeat words from the title.

§  Do not mix common and Latin names of species.

§  Explain better the novelty of your study.

§  Some parts of the Results section are in fact a repetition of Materials and methods and should be deleted (e.g. lines 98-103).

§  Avoid references in the Results section. These parts should be moved to the Discussion.

§  Figures 9 and 10: what statistical test was used for mean comparison?

§  Conclusions are missing

§  Latin names of species should be written in italics.

§  How did you obtain the plant material?

§  Name of cultivars should be written in inverted commas (‘’).

§  References lack doi numbers

The manuscript can be accepted for publication after incorporating all the necessary changes.

Comments on the Quality of English Language

English is fine
